# An artificial intelligence-based model for optimal conjunctive operation of surface and groundwater resources

Saeid Akbarifard [1,2] ✉, Mohamad Reza Madadi [3] ✉ & Mohammad Zounemat-Kermani[4] ✉

A hybrid simulation-optimization model is proposed for the optimal conjunctive operation of surface and groundwater resources. This second-level model is created by finding and combining the best aspects of two resilient metaheuristics, the moth swarm algorithm and the symbiotic organization search algorithm, and then connecting the resulting algorithm to an artificial neural network simulator. For assessment of the developed model efficiency, its results are compared with two first-level simulation-optimization models. The comparisons reveal that the operation policies obtained by the developed second-level model can reliably supply more than 99% of the total demands in the study regions, indicating its superior efficiency compared to the two other first-level models. In addition, the highest sustainability index in the study regions belongs to the proposed model. Comparing the results of this research with those of other recent studies confirm the supremacy of the developed second-level model over several previously developed models.

Water resource management in arid and semiarid regions is a critical challenge for managers and decision-makers. Iran is one of these arid regions, has experienced a drastic increase in water scarcity in the last decades due to its climatic condition. This problem has also been exacerbated by the population growth and expansion of agricultural, urban, and industrial activities. In such a situation, excessive exploitation of groundwater resources has caused a sharp drop in the groundwater level in most parts of the country. Therefore, the preservation of groundwater resources has become critical in Iran. Conjunctive operation of surface and groundwater resources can be considered a sustainable solution for reducing the high pressure on groundwater resources and preserving such valuable resources[1]. It is a suitable alternative for imbalanced water resource distribution and related constraints in groundwater exploitation[2]. Conjunctive operation has several benefits such as preventing additional investment in surface water resources (e.g., constructing dams or designing water transmission systems with an over-optimal capacity) as well as

avoiding excessive pressure on groundwater resources. Despite its several advantages, the appropriate implementation of the conjunctive operation of water resources is a complex engineering problem. Solving such complex problems (finding the best operation scenarios) by classical methods is very difficult (and often impossible) due to their several limitations. These limitations have turned the attention of researchers towards more efficient methods to solve such complex problems. During the last decade, developing soft computing techniques that often search based on the initial population has made it possible to find globally near-optimal solutions to very complex problems. Here, the results of some of the latest reputed studies on the application of such robust techniques for the conjunctive operation of surface and groundwater resources are presented.

Ghordoyee Milan et al.[3] used two fuzzy optimization methods, including a fuzzy inference system and a linear fuzzy optimization model, for the conjunctive operation of surface and groundwater resources in the Astaneh-Kouchesfahan Plain. The results

[1]Department of Water Engineering, Faculty of Civil and Surveying Engineering, Graduate University of Advanced Technology, P.O. Box 76315116 Kerman, Iran. [2]Research and Technology Institute of Plant Production, Shahid Bahonar University of Kerman, Kerman, Iran. [3]Department of Water Engineering, Faculty of Agriculture, University of Jiroft, Jiroft, Iran. [4]Department of Water Engineering, Faculty of Agriculture, Shahid Bahonar University of Kerman, Kerman, Iran. ✉e-mail: s.akbarifard@kgut.ac.ir; madadi@ujiroft.ac.ir; zounemat@uk.ac.ir

demonstrated that the linear fuzzy optimization model with an average deficit of 14.6% was the superior model compared to the fuzzy inference system (22%). Seo et al.[4] used a fully distributed hydrologic model for conjunctive management of groundwater and surface water resources during a drought. The satisfactory results of this model showed that the proposed management model took a step toward the sustainable exploitation of groundwater resources during drought periods. Sepahvand et al.[5] used a simulation-optimization model for conjunctive management of surface and groundwater resources of the Gavkhuni basin to increase the overall net benefit of agriculture and reduce irrigation water shortages. They simulated the interactions of surface and groundwater resources by a genetic programming model. Then, this simulation model was integrated with a multi-objective genetic algorithm as the optimization model. The findings showed that in wet, normal, and dry years, the net benefit was maximized by 38.19%, 59.37%, and 45%, respectively, compared to the non-optimized condition. Zeinali et al.[6] investigated the performance of non-dominated sorting genetic algorithm-II in optimal conjunctive operation of surface and groundwater resources in southwestern Iran. The results indicated that this optimization algorithm increased the reliability of the demands supply and reduced the over-exploitation of groundwater resources. Afshar et al.[7] adopted cyclic and non-cyclic approaches to the conjunctive operation of groundwater and surface water resources. The results revealed that the groundwater sustainability index in the cyclic conjunctive operation strategy was improved by more than 27% compared to the non-cyclic conjunctive operation strategy. Mirzaie et al.[8] studied the capability of the fuzzy multi-objective particle swarm optimization model in the conjunctive operation of groundwater and purified wastewater under uncertain conditions. In this study, three objective functions, namely maximizing economic profit, minimizing fertilizer consumption, and minimizing the withdrawal of groundwater resources, were evaluated in the Varamin irrigation network in Iran. The results indicated that the developed model raised the net benefit by 16% without increasing the cultivated area. Moreover, groundwater exploitation decreased by allocating a greater amount of recycled wastewater in total water consumption. By combining the system dynamics technique and the Nash bargaining theory, Naghdi et al.[9] optimized the conjunctive allocation of surface and groundwater resources in industrial, drinking, agricultural, and environmental sectors in the Najaf-Abad sub-basin. In this study, the groundwater extraction was kept to a minimum and the water supply was maximized using the non-dominated sorting genetic algorithm-II optimization algorithm. Based on the results, the water level of the aquifer optimally decreased during the study period. Arya Azar et al.[10] integrated the whale optimization algorithm and the firefly algorithm with the group method of data handling and least squares support vector machine to optimally allocate surface and groundwater resources in Marvdasht, south of Iran. The results indicated that the groundwater level increased by about 0.4 and 0.55 m using the whale optimization algorithm and firefly algorithm, respectively. Moreover, the firefly algorithm supplied about 61% of the water demands in the worst scenario for surface water resources, while this value was 52% using the whale optimization algorithm. Askari Fard et al.[11] developed an automated operating system, called centralized model predictive control, for the conjunctive operation of surface and groundwater resources in central Iran. They reported that by implementing the centralized model predictive control, the water extraction from the aquifer decreased by 16% after one year. Sondermann and Oliveira[12] used the WEI+ index to manage surface and groundwater resources in the Tagus River basin. They mentioned the advantages of this index for defining water scarcity levels compared to other indices. Calculating the WEI+ index in this basin indicated severe water stress conditions in most districts during the summer. Khosravi et al.[13] used conditional operation rules based on decision trees for the conjunctive use of groundwater and surface water resources. The objective function was defined as minimizing the water deficit in various river flow time series. The results revealed that the conditional operation rules reduced relative absolute error by 39% (minimum) and 71% (maximum) compared to a single linear regression. Khosravi et al.[14] compared the cyclic storage system and the standard conjunctive use method for the conjunctive operation of surface and groundwater resources. They employed a multi-objective mixed-integer nonlinear optimization model using the ε-constraint method for water allocation to irrigated agriculture and the energy required for groundwater pumping. Based on the findings, the cyclic storage operation strategy significantly improved the sustainability index compared to the standard conjunctive use strategy. Osorio Olivos et al.[15] proposed a framework for water allocation with RUBEM, MODFLOW, and PYWR hydrologic models. They used this framework in a basin of Sao Paulo and reported good adherence with the water balance patterns and the differences in demand attendance. Moghadam et al.[16] investigated the effects of climate change on the conjunctive operation of surface and groundwater resources with a real case study. They utilized the results of IHACRES (for surface water simulation) and MODFLOW (for groundwater simulation) to develop a conjunctive operation model with the WEAP model. They evaluated several climate change scenarios with different levels of future water demands with WEAP and successfully analyzed the annual deficits in future agricultural water supply. Moeini and Sarhadi[17] proposed a cyclic storage approach for the optimal conjunctive operation of surface and groundwater systems at the ZarrinehRoud basin. To solve this problem, they evaluated the performance of several mathematical and metaheuristics models including nonlinear programming, gravitational search algorithm, artificial bee colony algorithm, particle swarm optimization algorithm, and genetic algorithm in a real surface-groundwater system. They reported that the water demands were fully satisfied using the proposed methods. Although the nonlinear programming method decreased the operating cost, it extremely increased the computational time. Kayhomayoon et al.[18] proposed an approach for the conjunctive use of surface and groundwater resources. They used the MODFLOW and whale optimization algorithm to simulate the groundwater level and optimize the conjunctive use of water systems, respectively. Then they exported the results of these models to the least squares support vector machine model to predict the amounts of water deficits in the study period. They claimed that the developed model could accurately predict the water deficits as well, and the groundwater level could be increased by 0.7 m during the study period by employing the developed model. Shafa et al.[19] developed a multi-objective simulation-optimization platform for the optimal exploitation of surface and groundwater resources. Accordingly, they used HEC-HMS for flood routing and artificial recharge and the non-dominated sorting genetic algorithm for the optimization of cropping pattern and irrigation demand for an artificial recharge system. They documented that the developed model could decrease water consumption by 50%. Jain et al.[20] applied a multi-objective model for the optimal allocation of surface and groundwater resources with three objectives of maximizing the crop net return and the aquifer recharge and minimizing the water deficit in a real case study in India. Three optimization algorithms of particle swarm algorithm, genetic algorithm, and marine predators' algorithm, were used in the model. They introduced the marine predators' algorithm as the superior algorithm in achieving the best Pareto front, and it significantly increased the crop net return. Kalhori et al.[21] employed a multi-objective invasive weed optimization algorithm for optimal water allocation to drinking, industry, and agricultural demands under different climate change scenarios. They reported that the mentioned model could effectively reduce failure periods and allocate water resources to different consumption sectors.

This research aims to develop an efficient simulation-optimization model based on the hybridizing of three robust artificial intelligence

**Table 1 | Statistical indices of the best ANN model**

| Regions | Inputs | $R^2$ | RMSE | MAE | MSE | NMSE | MAPE |
|---|---|---|---|---|---|---|---|
| Baft | $Ev_{(t)}, Pr_{(t)}, Hy_{(t)}, De_{(t)}, G_{(t)}, Lev\_G_{(t-1)}$ | 0.9989 | 0.1256 | 0.064 | 0.0158 | 0.0011 | 0.0032 |
| Rabor | | 0.9994 | 0.088 | 0.0556 | 0.0077 | 0.0006 | 0.0028 |
| Jiroft | | 0.9978 | 0.2058 | 0.1159 | 0.0424 | 0.0022 | 0.0187 |

**Table 2 | Objective function values and total demand deficit obtained by the developed models**

| Method | Objective function | Baft | | Rabor | | Jiroft | |
|---|---|---|---|---|---|---|---|
| | | Water supply (%) | Deficit (MCM) | Water supply (%) | Deficit (MCM) | Water supply (%) | Deficit (MCM) |
| SOS-MSA-ANN | **399.28** | **99.49** | **1.21** | **99.50** | **2.82** | **99.73** | **3.17** |
| MSA-ANN | 544.68 | 96.54 | 8.17 | 96.49 | 19.92 | 99.39 | 7.24 |
| SOS-ANN | 769.27 | 96.57 | 8.10 | 98.23 | 10.04 | 98.89 | 13.12 |

Bold values indicate better performance of the corresponding model.

methods for the optimal allocation of surface and groundwater resources in the Halilrood basin. Accordingly, first, a sensitivity analysis is performed on the robust moth swarm algorithm structure to identify its most efficient operators. Then, these operators are exported to the search process of the symbiotic organism search algorithm (to strengthen this algorithm) to form a hybrid optimizer. Finally, the artificial neural network as a simulator is linked with the developed optimizer to obtain a second-level simulation-optimization model, SOS-MSA-ANN, which can minimize the water deficit in different sectors of the Halilrood basin, subject to constraints on groundwater drawdown. The main novelties of this research are: (1) structural sensitivity analysis of moth swarm algorithm and identifying the main operators of this algorithm (celestial navigation, transverse orientation, and Lévy-mutation), (2) developing the robust hybrid symbiotic organism search- moth swarm algorithm by employing the powerful operators of the latter into the search engine of the former, (3) developing the second-level hybrid simulation-optimization model, SOS-MSA-ANN, for the conjunctive operation of surface and groundwater resources, (4) implementing the developed simulation-optimization model in a real case study lacking sustainable water resource management protocols.

## Results
### Groundwater level estimation by ANN simulation model
The RBF-ANN model was adopted to estimate the groundwater level in the studied areas. Table 1 represents the results of the superior neural network model in estimating the groundwater level. Evidently, the values of the determination coefficient ($R^2$) in all regions are more than 0.99 while the values of error indices are very small, e.g., less than 0.05 for MAPE, indicating the significant performance of the selected neural network model in groundwater level estimation. The high performance of neural network in estimating the groundwater level in all the studied areas (Baft, Rabor, and Jiroft) indicates that it is a reliable simulation model for linking with the optimization models. Supplementary Fig. 1 illustrates the distribution diagrams of observed (measured data) and simulated values of groundwater level in three regions of Baft, Rabor, and Jiroft. In this figure, the values of the horizontal and vertical axes denote the elevation of groundwater level in the studied area (MASL). Almost all the points are concentrated on the 45-degree line. This implies that there is a great agreement between the simulated values and the observed data. This figure indicates that the neural network model has high efficiency in estimating the groundwater level in all three regions of Baft, Rabor, and Jiroft.

Due to the significant performance of the artificial neural network model, as proved by Table 1 and Supplementary Fig. 1, it was employed in the simulation-optimization model as a powerful simulator for the

**Table 3 | Values of evaluation criteria of the developed models**

| Regions | Model | Reliability | Resiliency | Vulnerability | Sustainability |
|---|---|---|---|---|---|
| Baft | **SOS-MSA-ANN** | 96.41 | 87.5 | 15.59 | **89.30** |
| | **MSA-ANN** | 89.24 | 74 | 55.08 | 66.69 |
| | **SOS-ANN** | 91.03 | 77.27 | 72.84 | 57.59 |
| Rabor | **SOS-MSA-ANN** | 96.86 | 70.37 | 20.35 | **81.58** |
| | **MSA-ANN** | 91.03 | 60 | 35.62 | 70.58 |
| | **SOS-ANN** | 95.96 | 78.72 | 33.55 | 79.48 |
| Jiroft | **SOS-MSA-ANN** | 98.21 | 100 | 6.9 | **97.06** |
| | **MSA-ANN** | 95.07 | 100 | 27.28 | 88.42 |
| | **SOS-ANN** | 91.93 | 78.95 | 41.56 | 75.13 |

Bold values indicate better performance of the corresponding model.

conjunctive operation of surface and groundwater resources in the Halilrood basin.

### Evaluation of the simulation-optimization model outputs
In this study, the objective function was minimizing the total deficit in the study period. Here, the results of the SOS-MSA-ANN model in comparison with two other developed models of SOS-ANN and MSA-ANN, were presented for the conjunctive operation of surface and groundwater resources in the Halilrood basin. Tables 2 and 3 demonstrate the best values of the objective function and the performance indicators of water systems in Baft, Rabor, and Jiroft, obtained by the developed models after 1000 iterations. As presented in Table 2, the objective function value of the developed hybrid SOS-MSA-ANN model in the problem was equal to 399.28, indicating the higher capability of this model compared to the neural network moth swarm algorithm (objective function=544.68) and neural network symbiotic organism search algorithm (objective function=769.27). The SOS-MSA-ANN supplied 99.49%, 99.50%, and 99.73% of the demands in Baft, Rabor, and Jiroft, respectively. The neural network moth swarm algorithm and neural network symbiotic organism search algorithm models with the demand supply of (96.54%, 96.49%, and 99.39%) and (96.57%, 98.23%, and 98.89%) occupied the next ranks. The better performance of the SOS-MSA-ANN indicates that it could better manage (reduce) the total deficits in the study period compared to the other two models.

Table 3 shows the values of evaluation criteria (reliability, resilience, vulnerability, and sustainability indices) of the developed models in the conjunctive operation of surface and groundwater

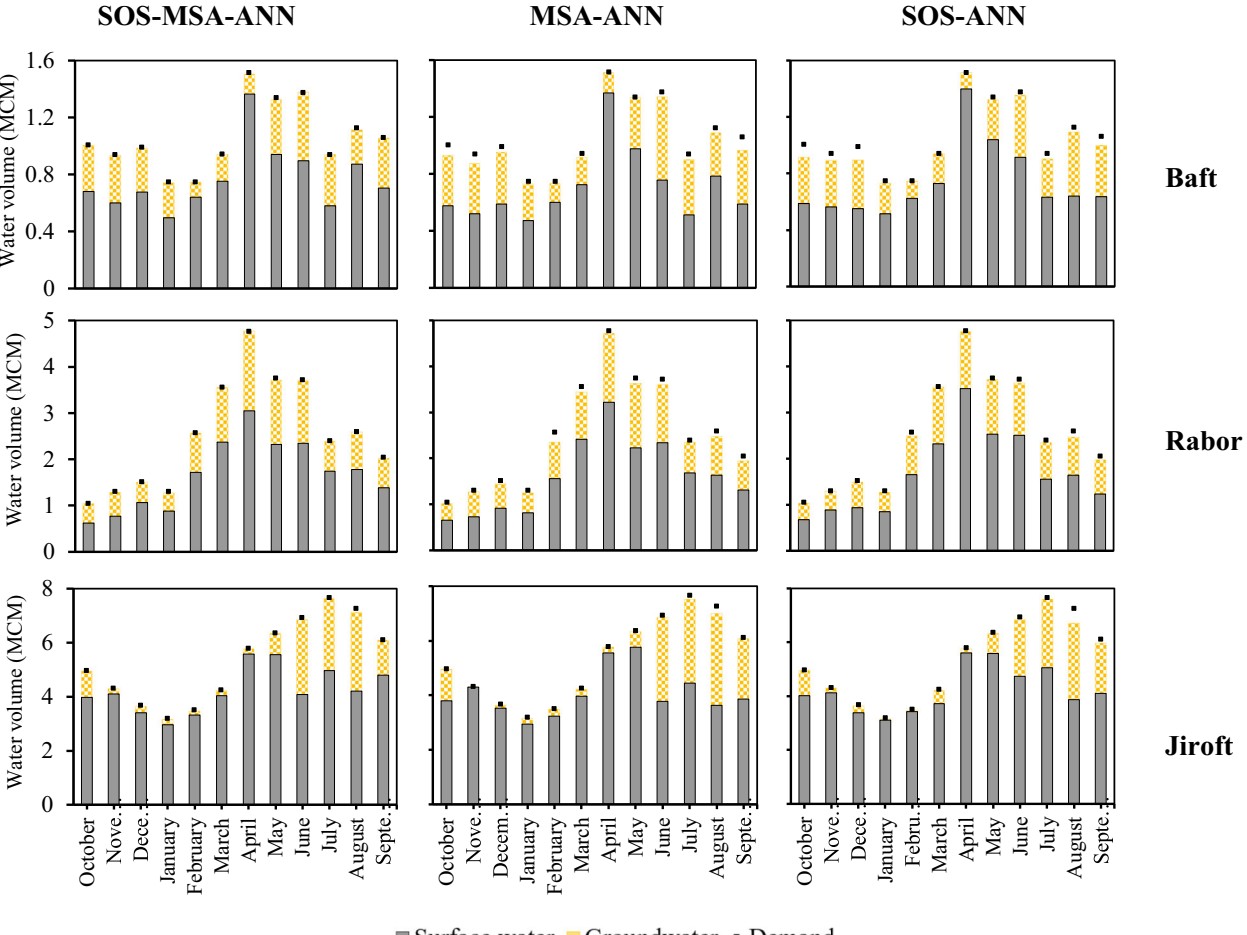

**Fig. 1 | Monthly average allocation of surface and groundwater resources to demands in Baft, Rabor, and Jiroft by the developed models (study period: 2001–2019).** First, second and third rows relate to the results of the models in the Baft, Rabor, and Jiroft regions, respectively. SOS-MSA-ANN is the second-level simulation-optimization model produced by the hybridization of two metaheuristics of symbiotic organism search algorithm and moth swarm algorithm with the artificial neural network. MSA-ANN and SOS-ANN denote the first-level hybrid artificial neural networks with moth swarm and symbiotic organism search algorithms, respectively. The histograms show the contribution of surface and groundwater resources in meeting the demands (black points) of each region. In some months, the total water supplied from the "surface water" and "groundwater" sources is almost equal to the total water "demands". This means that the utilized optimization model has been able to supply almost all the water demands of that region in that month. Source data are provided as Source Data file.

resources in Baft, Rabor, and Jiroft. The sustainability index obtained by the SOS-neural network moth swarm algorithm model to supply total demands in Baft, Rabor, and Jiroft (89.30, 81.57, and 97.06) was significantly greater than the corresponding values of neural network moth swarm algorithm (66.69, 70.58, and 88.42) and neural network symbiotic organism search algorithm (57.59, 79.48, and 75.13). There is a similar trend for the other indicators. In other words, the comparison of all 4 indicators showed that the SOS-MSA-ANN was the superior model in the optimization of the conjunctive operation of surface and groundwater resources.

Supplementary Fig. 2 indicates the convergence rate of the second-level SOS-MSA-ANN model in comparison with the first-level MSA-ANN and the SOS-ANN models. The SOS-MSA-ANN produced the closest solutions to the optimal solution in the least number of iterations. It is seen that for an equal number of iterations in all three models (1000 iterations), the values of the solutions generated by SOS-MSA-ANN were far less than the neural network moth swarm algorithm and neural network symbiotic organism search algorithm, respectively.

Figure 1 indicates the monthly average allocation of surface and groundwater resources to the demands in Baft, Rabor, and Jiroft by the developed models in the study period (2001–2019). By comparing the results of water allocation, it is clear that the SOS-MSA-ANN supplied

nearly 100% of the total demands in all months, without any significant deficit in a certain month or region. However, the other models were not successful in some months. For example, in the Baft region, the MSA-ANN and SOS-ANN could not supply the demands satisfactorily from October to December, July, and September. This revealed the high efficiency of the SOS-MSA-ANN model in the optimal operation of this complex water system.

Figure 2 shows the simulated groundwater level by the developed models resulting from the conjunctive operation scenarios versus the observed values in the same period in Baft, Rabor, and Jiroft. The black line with triangular markers (SOS-MSA-ANN) is the highest curve, and the blue line with circular markers (real-condition operation) is the lowest curve of this figure. It means that, the SOS-MSA-ANN followed by MSA-ANN and SOS-ANN kept the groundwater level at a higher elevation compared to the real condition. In other words, the groundwater level drop in the non-optimal condition was about 2 m more than the optimal condition in all the studied regions, showing that a large volume of groundwater was preserved by the optimal management scenarios. Therefore, it can be concluded that all the developed models could optimize the conjunctive operation of surface and groundwater resources in the Halilrood basin and could supply the demands with high reliability. In addition, the resulting scenarios were effective in the partial balance of groundwater, so they

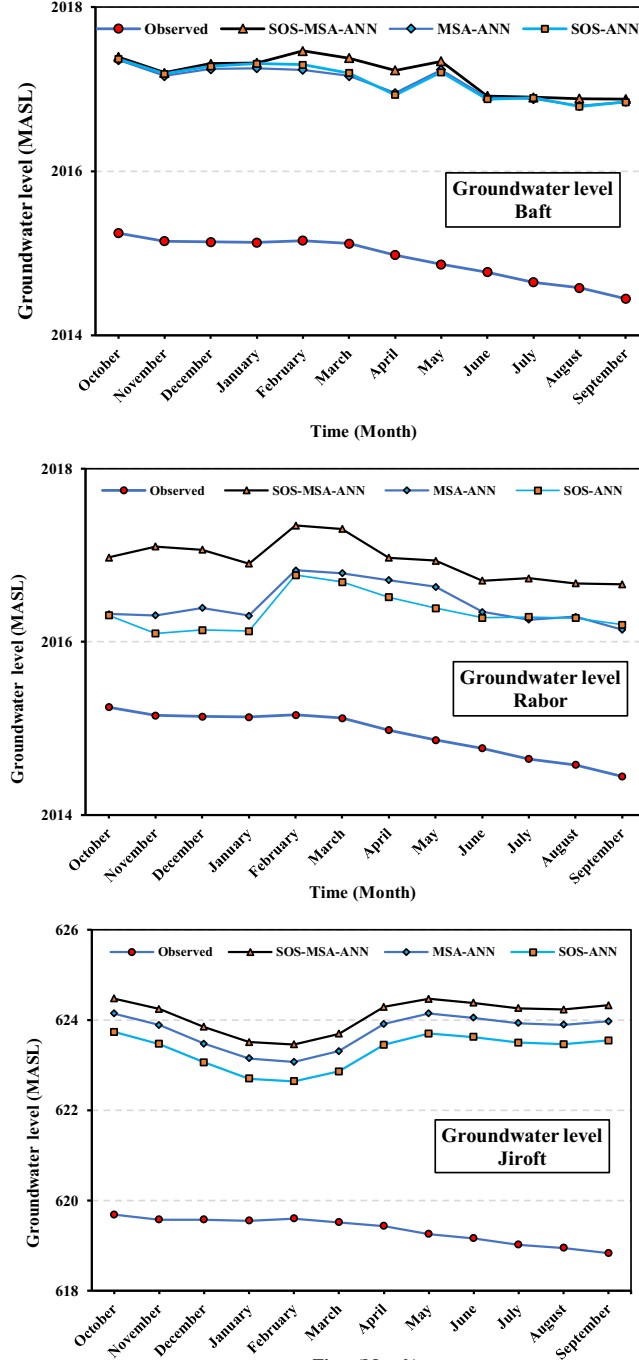

**Fig. 2 | Monthly average groundwater level in the three regions simulated by the developed hybrid models versus the actual condition in the study period (2001–2019).** The top figure shows the variations of observed (measured) groundwater level compared to the simulated by the developed models (SOS-MSA-ANN, MSA-ANN, and SOS-ANN) in the Baft region. The middle and below figures show the same items for the Rabor and Jiroft regions, respectively. Vertical axis demonstrates the groundwater level by the meter above sea level as the datum. Source data are provided as Source Data file.

balanced the groundwater level and maintained its sustainability in the study areas during the 19-year period.

Supplementary Fig. 3 indicates the monthly deficits in the Halilrood basin resulting from the conjunctive operation of surface and groundwater resources by the developed models. In this figure, the blue bars show the values of the deficits in real-condition operation.

**Table 4 | Comparison of different hybrid models in conjunctive operation of water resources**

| Criteria | Reference | Model | Result |
|---|---|---|---|
| Water supply (%) | Zeinali et al.[6] | NSGA-II-WEAP-MODFLOW | 97.85% |
| | Arya Azar et al.[10] | FA-GMDH-LS-SVM | 87.1% |
| | | WOA-GMDH-LS-SVM | 84.4% |
| | Current study | MSA-ANN | 97.5% |
| | | SOS-ANN | 97.9% |
| | | **SOS-MSA-ANN** | **99.6%** |
| Sustainability index (%) | Khosravi et al.[14] | CS Model | 86% |
| | | SCU Model | 70% |
| | Current study | SOS-ANN | 70.73% |
| | | MSA-ANN | 75.23% |
| | | **SOS-MSA-ANN** | **89.3%** |
| Groundwater level rising (%) | Naghdi et al.[9] | NSGA-II and Nash bargaining theory coupled model | 30% |
| | Current study | SOS-ANN | 56.33% |
| | | MSA-ANN | 62.21% |
| | | **SOS-MSA-ANN** | **64.68%** |

Bold values indicate the best model in terms of each criteria.

During all operational months, the values of deficits obtained by SOS-MSA-ANN were much less than those of the two other models, indicating the high performance of this model in supplying the demands and optimal conjunctive operation of surface and groundwater resources. It reduced the deficits by 1.21, 2.82, and 3.17 Mm³ in Baft, Rabor, and Jiroft, respectively, during the 19-year operational period. The corresponding values for the neural network moth swarm algorithm were 8.17, 19.92, and 7.24 Mm³ which placed it at the next rank beside the neural network symbiotic organism search algorithm.

## Discussion

This study attempted to develop a hybrid simulation-optimization model to supply the water demands in the Halilrood basin in Iran. Here, the results of this study are compared with those of some similar studies on the conjunctive use of water resources. As seen in Table 4, three criteria of (i) the percentage of total demands supplied by the utilized model, (ii) the sustainability index of the studied water resources system optimized by the models, and (iii) the capability of utilized models in raising the groundwater level compared to the actual condition, were employed to compare the performance of utilized models. In terms of water supply, the MSA-ANN, SOS-ANN, and SOS-MSA-ANN models (the developed models of the present study) could successfully supply 97.5%, 97.9%, and 99.6% of the total demands in the studied basin. Zeinali et al.[6] developed a hybrid model by linking the MODFLOW, WEAP, and NSGA-II which could supply 97.85% of the demands. The performance of their model was comparable with the performance of two first-level models of the present study (MSA-ANN and SOS-ANN), but its performance is far from the performance of the second-level model of SOS-MSA-ANN. Furthermore, two hybrid metaheuristics models of FA-GMDH-LS-SVM and WOA-GMDH-LS-SVM proposed by Arya Azar et al.[10] with water supply of 87.1% and 84.4%, could not provide significant results in meeting the demands of the studied area. Therefore, among the six implemented methods for the conjunctive use of water resources systems, the SOS-MSA-ANN was the superior model in terms of water supply. It should be noted that each of these models was developed for a specific area and this issue should be considered in the comparisons. In terms of the sustainability index, the best results were also obtained by SOS-MSA-ANN (89.3%), followed by the CS model (86%) developed by Khosravi et al.[14]. The next rank was devoted to the MSA-ANN model with SI =

75.23%. The performance of the SOS-ANN model was somewhat similar to that of the SCU model, both of which were about 70%.

By optimizing the conjunctive operation of water resources systems to sustain the groundwater resources, it is anticipated that the groundwater level will increase during the study period compared to the non-optimal condition. Accordingly, Naghdi et al.[9] claimed that by combining the NSGA-II and Nash bargaining theory coupled model, they could increase the groundwater level by about 30% (compared to the actual condition), while the increase of groundwater level by the SOS-ANN and MSA-ANN models was 56.33% and 62.21%, respectively, indicating the superiority of the developed models of the present study. The highest increase in groundwater level was obtained by the SOS-MSA-ANN model, 89.3%, which indicates its impressive capability of sustainable exploitation of groundwater resources.

For a better understanding, the results of the three models developed in this study in meeting the demands of the whole basin were compared with each other in Fig. 3. In this figure, the water withdrawal from the groundwater resources is indicated by blue dotted bars and the withdrawal from the surface water (reservoirs) is demonstrated by purple horizontal stripes bars. The values of withdrawals are expressed as a percentage of the total surface and groundwater usage by a yearly scale. For example, in 2016, the SOS-MSA-ANN model supplied 99% of the total demands in the whole basin, of which 77% was supplied by the surface water, and 22% was supplied by the groundwater (the values on the bars are rounded to the nearest integer number). The surface water (the stored water in the dam reservoirs) is the main source of water supply in the whole basin. While the SOS-ANN and MSA-ANN could not meet all the demands in most years, the SOS-MSA-ANN met nearly 100% of the demands in most years of the study period. In addition, the next important point that can be found from this figure is that in the management scenario produced by SOS-MSA-ANN, compared to the other two models, less water withdrawal from the groundwater source (aquifer) was performed. This means that in the management scenario produced by this model, while a higher percentage of the basin's demands has been met, the groundwater level has also become more stable; in other words, the sustainable exploitation of groundwater resources has taken place.

## Methods
### Study area
The Hamun-Jazmurian basin with an area of 69375 km² in southeastern Iran is a part of the central desert basin of Iran and is geographically located at 56° 15′–61° 23′ E longitude and 26° 28′–29° 30′ N latitude. The Halilrood basin is located in the western part of the Hamun-Jazmurian basin within the Kerman Province (Fig. 4) and covers an area of about 7224 km².

Halilrood is the greatest river in terms of discharge (with an average annual discharge of 7.68 m³/sec) in Kerman Province and one of the main water sources for the Jazmurian wetland[22]. The Halilrood flow is used for energy generation (by the Jiroft hydropower dam), as well as, agricultural, domestic, industrial, and environmental uses. The basin elevation varies from 1391 m to 4359 meter above sea level (MASL). In this basin, rainfed agriculture is limited and irrigated agriculture is mainly established in the vicinity of rivers, qanats, and springs. According to the Köppen–Geiger climate classification system, the Halilrood basin has an arid desert climate with hot summers. While the average annual temperature is 13 °C, the daily average maximum temperature approaches 40 °C. The long-term average annual rainfall is less than 225 mm, most of which is received between January and May, while precipitation is negligible from June to December. The annual potential evapotranspiration (PET) ranges from 2039 mm to 2569 mm. Five hydrometric stations, 9 climate stations, and one synoptic station are in operation, providing daily climate data since 1979 and discharge data since 1993. In the northern part of the basin, the Baft and Safarood dams were constructed for agricultural,

drinking and industrial purposes. At the basin outlet, Jiroft Dam was built for domestic, industrial, and agricultural uses, power generation and flood control purposes. In recent decades, besides surface water, the extraction of groundwater has been widely performed to combat severe droughts. Accordingly, the water needed for domestic, industrial, and agricultural purposes is supplied by a combination of surface and groundwater resources (wells, qanats, dams, and springs).

### Utilized data
In this study, 223-month time-series data, from 2001 to 2019, were used for the conjunctive operation of surface and groundwater resources in the Halilrood basin to minimize the deficits in meeting the demands. The utilized data include precipitation, evaporation, river flow, groundwater level, and total demands in the Baft, Rabor, and Jiroft regions (Supplementary Table 1).

### Artificial neural network (ANN) simulation model
As mentioned in the previous sections, the multi-layer perceptron (MLP) artificial neural network model was used to simulate the groundwater level in the studied regions. For this purpose, the measured monthly data of precipitation ($Pr$) in the current time step, evaporation ($Ev$) in the current time step, river flow ($Hy$) in the current time step, water demand in each region ($De$) in the current time step, aquifer exploitation ($G$) in the current time step, piezometric groundwater level in each region in the previous time step (as the initial conditions) and piezometric groundwater level in the adjacent regions in the previous time step (as the boundary conditions) were used as the neural network model inputs, and the groundwater level in the current time step was considered as the model output. Therefore, there are a total of seven inputs and one output for the neural network model in each region. Different combinations of input parameters and different structures were designed and tested to find the appropriate structure for the neural network model, and finally, the best model was selected to estimate the monthly groundwater level. Accordingly, the best neural network model had three hidden layers with a structure of (10-15-10) and one output layer. The radial basis function (RBF) was determined as the middle layers transfer function. The Purelin was specified as the output layers transfer function. In the training process, the initial weights were randomly assigned, each network was trained with several iterations to prevent the network from being trapped in local minima, and the best result was considered as the criterion of action. The mean square error (MSE) was used as the performance function and the Levenberg-Marquardt algorithm was employed as the training function. The results of simulations were compared with the observed data of groundwater level to evaluate the reliability of the developed ANN model. All the programs, as provided in Supplementary Code 1, were coded in MATLAB (R2018b).

### Symbiotic organisms search (SOS) algorithm
Symbiotic organisms search (SOS) is a meta-heuristic optimization algorithm, that simulates the interactive behaviors of organisms[23]. Organisms rarely live in isolation due to relying on other species to live and even survive. This trust-based relationship is known as symbiosis. The SOS algorithm simulates interactions in the relationship between two species, in a way that one species searches to find the most suitable organism. Like other population-based algorithms, the SOS algorithm repeatedly generates a population of candidates to find regions as optimal solutions in the whole solution space. This population is equivalent to the optimal release scenarios in each period, in the problem of optimal operation of the reservoirs. The SOS algorithm starts with an initial population called the ecosystem. In the initial ecosystem, a group of organisms (decision variables) is randomly generated in the search space. Each organism as a candidate for the problem solution, which is related to a certain degree of fitness, indicates the degree of adaptation to the desired target (value of the

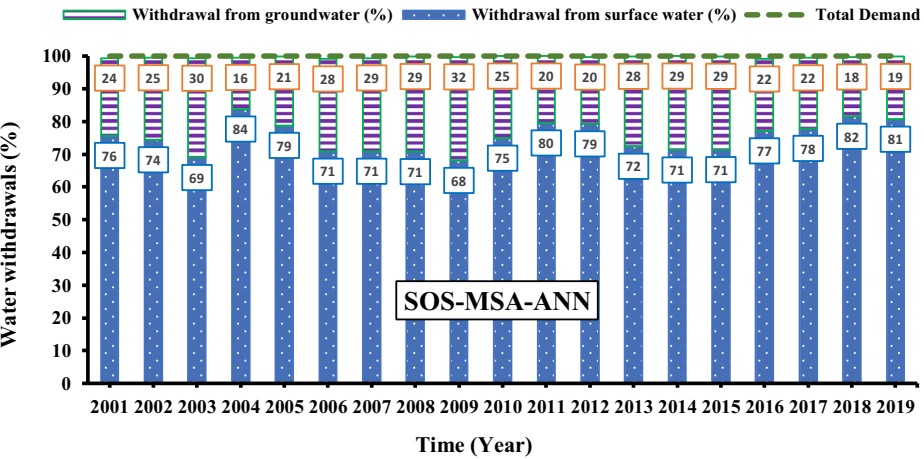

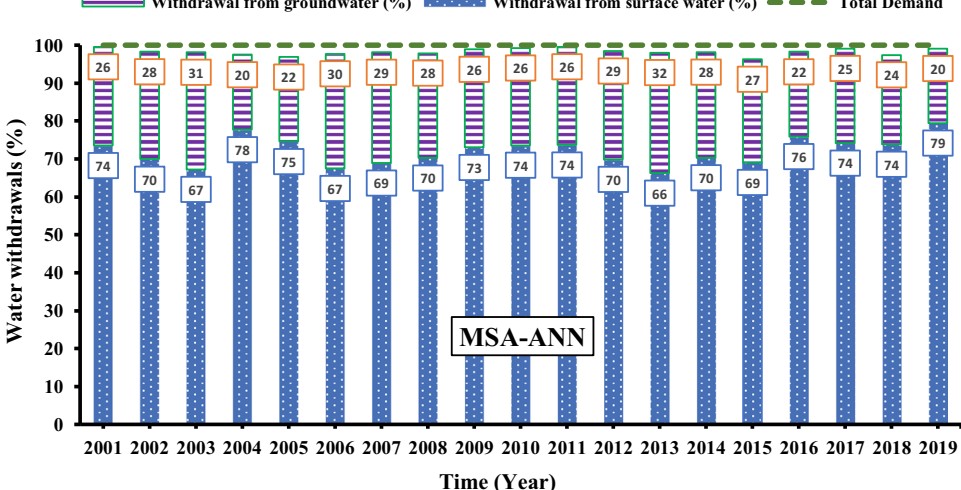

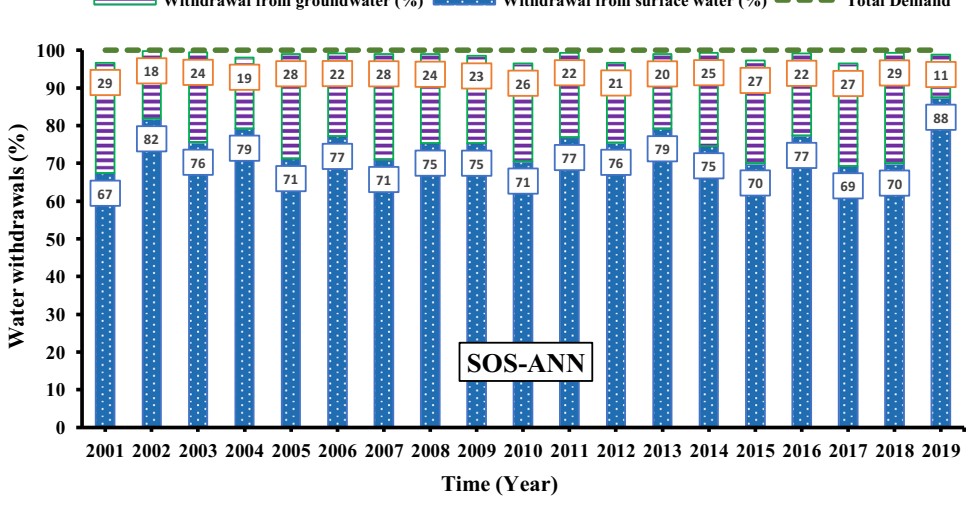

**Fig. 3 | Optimized yearly withdrawal from each water resources (in percent) by the utilized models versus the total demands.** The histograms show the optimized annual withdrawal from the surface and groundwater resources in the whole Halilrood basin (including all the regions) in the study period (2001–2019). The top, middle, and bottom figures are the optimized withdrawal simulated by the **SOS-MSA-ANN, MSA-ANN**, and **SOS-ANN**, respectively. The value inside the box denotes the withdrawal from each resource by percent. The horizontal green dashline shows the total demands in the whole basin. In some years, the developed models could meet the total (100%) downstream demands of the whole basin by optimizing water withdrawal from the surface and groundwater resources. Source data are provided as Source Data file.

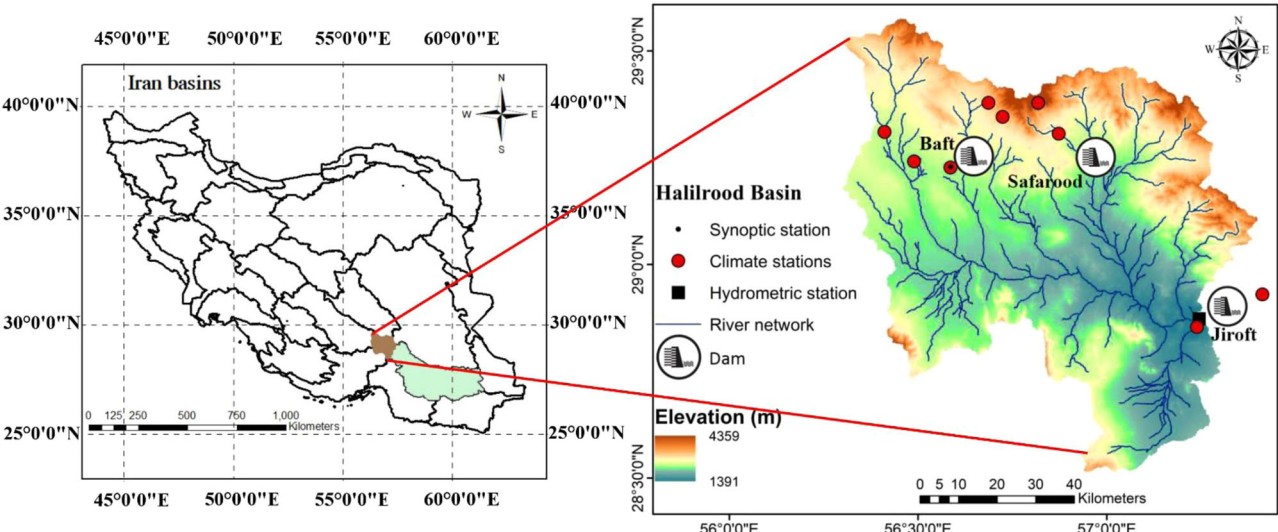

**Fig. 4 | The Halilrood basin in Iran; the main river tributaries and the location of dams and climatic/hydrometric stations are shown in the zoomed view.** The left map demonstrates the location of large Hamun-Jazmurian basin (green area) in Iran and the location of Halilrood Basin (brown area) inside it. The right map shows the zoomed view of the Halilrood Basin with the river tributaries and the location of dam reservoirs in the study area.

objective function). Almost all meta-heuristic algorithms use a substitution process at each iteration to solve the problem and generate a new solution for the next iteration. In SOS, the generation of a new solution is governed by mimicking the biological interaction between two organisms in the ecosystem. Three phases of mutualism, commensalism, and parasitism, which are like the biological interaction model in the real world, are considered. The identity of each interaction is defined based on the type of interaction. In this way, the two-way profit shows the mutualism phase, the one-sided profit represents the commensalism phase, and the profit of one side and the loss of the other side indicate the parasitic phase. In all phases, each organism randomly interacts with another organism. This process continues until the process termination criterion (reaching the maximum number of iterations) is met. The general process of the algorithm is as follows[23]:

Initialization → Iteration → Mutualism phase → Commensalism phase → Parasitism phase → Process termination after reaching the maximum number of iterations.

### Moth swarm algorithm (MSA)

The moth swarm algorithm (MSA) was inspired by the behavior of moths in nature[24]. During the day, moths strive to avoid predators, while at night, they use celestial navigation to find food sources. They fly in a straight line over long distances and act as a compass at a constant angle towards a light source like the moonlight[24]. To prevent premature convergence, i.e., being trapped in local solutions, and achieving the global optimum, the moth swarm algorithm includes three groups of moths, namely pathfinders, prospectors, and onlookers. The associative learning mechanisms with instantaneous memory, and the population division for Lévy-mutation were employed in the MSA to enhance its exploitation and exploration capabilities. Gaussian walks and adaptive spiral motion are other mechanisms of the MSA that increase the convergence rate, flexibility against local optimum problems, and the capability to face large-scale and complex problems[25].

As mentioned earlier, by performing a structural sensitivity analysis on the robust MSA algorithm, the most effective operators of this algorithm were identified and then used to improve the SOS algorithm, for generating the hybrid SOS-MSA algorithm. In the following section, the procedure for generating the hybrid SOS-MSA algorithm is explained in more detail.

### Structural sensitivity analysis of MSA

By performing a sensitivity analysis on the structure of the MSA and the way its operators work, it is possible to identify the strong and effective operators of the MSA algorithm and use them to strengthen and upgrade other algorithms and generate new hybrid algorithms. For this purpose, a brief explanation about each of these operators is first presented, and then by changing or removing each of these operators one by one, the strongest ones which are responsible for the high capability of the MSA algorithm are identified and then added to the SOS algorithm to generate the hybrid SOS-MSA.

### Lévy-mutation mechanism

Pathfinder moths are required to explore less crowded areas to avoid premature convergence (being trapped in local optima),. Pathfinders update their position by interacting with each other (crossover), flying over long distances (Lévy-mutation), and performing adaptive displacement using Lévy-mutation mechanism. Lévy-mutation is a random process based on α-stable distribution with the ability to move over long distances. The α-stable distribution is strongly associated with the probability density function, statistical fractal theory, and inhomogeneous diffusion.

### Transverse orientation mechanism

Prospector moths, which have the highest luminescence intensity after pathfinder moths, travel the spiral path by conical logarithmic spirals. In the transverse orientation phase, each prospector moth updates its position based on the spiral flight path (logarithmic flight). In moth swarm algorithm, the type of each moth changes dynamically, e.g., the number of prospector moths decreases, and the number of onlooker moths increases at different iterations. Moreover, if the prospector has a higher luminescence intensity than the pathfinder, it can become a pathfinder.

### Celestial navigation mechanism

Onlooker moths that have the lowest luminescence intensity move directly toward the best solution (moonlight). Onlookers focus on important points in the search space to search more effectively. In moth swarm algorithm, onlooker moths are divided into two categories:

(I) Gaussian walks: The first group of onlookers uses the Gaussian random distribution to move toward the moonlight due to the ability

of this distribution to limit random sample distribution. Gaussian walks limit variation in subsequent population positions.

(II) Associative learning mechanism with immediate memory: Immediate memory is a type of memory used for transferring information from one generation to the next generation in algorithms. Moth behaviors are strongly influenced by associative learning with immediate memory. The second group of onlooker moths are drawn to the moonlight using associative learning with immediate memory. Immediate memory is initialized with Gaussian continuous uniform distribution from $x_i^t - x_i^{min}$ to $x_i^{max} - x_i^t$.

In this study, different structures were obtained by modifying or removing the main operators of the MSA. The investigated structures are explained In the following.

Structure 1: The MSA was examined without any changes in the structure.

Structure 2: The Cauchy distribution (instead of the Lévy distribution) was used in the Lévy-mutation phase of the original MSA. The Cauchy distribution, $q \sim cauchy(\sigma, \mu)$, was defined as follows:

$$f(q) = \frac{1}{\pi(\sigma^2 + (q - \mu)^2)} \rightarrow -\infty < q < \infty \tag{1}$$

Structure 3: The normal or Gaussian distribution (instead of the Lévy distribution) was used in the Lévy-mutation operator of the original MSA. The normal or Gaussian distribution $q \sim N(\mu, \sigma_G^2)$ with density function is presented as Eq. (2):

$$f(q) = \frac{1}{\sqrt{2\pi}\sigma_G} \exp\left(-\frac{(q - \mu)^2}{2\sigma_G^2}\right) \rightarrow -\infty < q < \infty \tag{2}$$

Structure 4: The Lévy-mutation process or mechanism, which controlled the movement of pathfinder moths in the search space, was removed from the original MSA algorithm.

Structure 5: The transverse orientation process, which uses the logarithmic spiral mechanism to move prospector propellers, was removed from the original MSA algorithm.

Structure 6: The Gaussian walks mechanism, which controlled the movement of onlooker moths, in the celestial navigation phase was removed from the original MSA algorithm.

Structure 7: The associative learning mechanism with the immediate memory was removed from the celestial navigation phase in the original MSA algorithm.

Structure 8: The Gaussian walks mechanism and the associative learning mechanism with the immediate memory, i.e., the celestial navigation phase, were simultaneously removed from the original MSA algorithm.

A benchmark problem was needed to investigate the effect of these 8 structures on the performance of the MSA algorithm and to identify its strongest and most effective operators. Accordingly, the four-reservoir benchmark problem, as the closest test problem to the problem of conjunctive operation of surface and groundwater resources, was used[26]. This multi-reservoir system consists of both series and parallel reservoirs in which the operation policy of the upstream reservoirs affects the downstream reservoirs. This benchmark has 46 decision variables, and the objective function is to maximize the profit of the system in 12 operating periods[26]. In this figure, $Q_i$ and $R_i$ represent the inflow and release from reservoir $i$, respectively.

The global optimal value in this problem is equal to 308.83, and the proximity of the objective function to this value is considered the reason for the superiority of an optimization model. Table 5 represents the results of eight MSA structures at 10 runs. In the overall ranking, Structure 8 (removing the celestial navigation phase) had the worst rank among the investigated structures. It means that the worst value of the objective function (268.151) was obtained by this structure. This indicates the dramatic (negative) impact of removing the celestial

navigation phase on the performance of the MSA algorithm. The next worst solutions were obtained by the MSA with Structures 6 and Structure 7 (removing the Gaussian walks mechanism and removing the associative learning mechanism with immediate memory, respectively). In other words, removing the celestial navigation phase, including the Gaussian walks mechanism, and the associative learning mechanism with immediate memory, dramatically reduced the performance of the MSA algorithm. Moreover, removing the Lévy-mutation and the transverse orientation phases reduced the algorithm's capability and caused entrapment in local optima, respectively.

Supplementary Fig. 4 provides the convergence rate to the optimal value in eight MSA structures for the benchmark problem. Evidently, Structures 4, 6, and 8, obtained by removing the Lévy-mutation and the celestial navigation phases, negatively impacted the algorithm convergence rate such that, in these cases, the algorithm was trapped in local optima in several steps.

## SOS-MSA optimization model
According to the sensitivity analysis of the structure of the moth swarm algorithm, it was observed that the Lévy-mutation, the celestial navigation, and the transverse orientation operators had a great effect on the strength of the MSA algorithm. Therefore, it can be concluded that these operators are the most effective parameters of the MSA algorithm that play a significant role in its performance. Thus, if these strong items are used in another optimization model, it is anticipated that the performance of that algorithm should also increase significantly. Accordingly, these operators were exported to the search engine of the symbiotic organism search algorithm to generate the hybrid SOS-MSA optimization algorithm.

## Objective function of the optimization model
The optimization model aims to meet the maximum demands or in other words, to minimize the difference between the allocated water and the demands as follows:

$$Minimize f = \sum_{i=1}^{3}\left[\sum_{t=1}^{T}\left(Demand_{i,t} - Supply_{i,t}\right)^2\right] + Penalty1_i \\ + Penalty2_i \tag{3}$$

where, $f$ is the objective function, $t$ is the time step counter (number of months in the planning period), $i$ is the area counter, $Demand_{i,t}$ is the net water demand in the $i$th region and $t$th month, $Supply_{i,t} = G_{i,t} + Re_{i,t}$ where $Supply_{i,t}$ is the amount of water allocated to the $i$th region in the $t$th month by the optimization model, $G_{i,t}$ is the groundwater withdrawal in the $i$th region and $t$th month, $Re_{i,t}$ is the surface water exploitation in the $i$th region and $t$th month, $Penalty1_i$ and $Penalty2_i$ are the penalty functions related to the volume of reservoirs (Eq. (12)) and groundwater level (Eq. (15)), respectively.

## Constraints of the optimization model
Constraints related to spillway and evaporation losses are applied in the form of the following relations:

$$Sp_{i,t} = \begin{cases} S_{i,t} - S_{max i} + S_{min i} & \text{if } S_{i,t} > (S_{max i} - S_{min i}) \\ 0 & \text{if } S_{i,t} \leq (S_{max i} - S_{min i}) \end{cases} \tag{4}$$

$$Loss_{i,t} = A_{i,t} \times Ev_{i,t} \tag{5}$$

$$A_{i,t} = a_i + b_i \times S_{i,t} + c_i \times S_{i,t}^2 \tag{6}$$

- In all stages of optimization of reservoirs operation, there must be a mass balance between the input and output values and the volume of the reservoir (continuity relationship). This is demonstrated

**Table 5 | Results of ten different runs on MSA structures in the benchmark four-reservoir system**

| Number of runs | Structure 1 | Structure 2 | Structure 3 | Structure 4 | Structure 5 | Structure 6 | Structure 7 | Structure 8 |
|---|---|---|---|---|---|---|---|---|
| 1 | 308.8324 | 303.2897 | 303.8548 | 281.9172 | 300.9973 | 283.0921 | 293.0915 | 289.9782 |
| 2 | 308.5169 | 302.1507 | 308.2725 | 281.8771 | 298.7954 | 279.8854 | 300.8698 | 282.4970 |
| 3 | 308.6384 | 307.1987 | 302.3948 | 284.3652 | 300.1170 | 273.2014 | 296.1268 | 279.8042 |
| 4 | 307.5226 | 304.9934 | 307.4276 | 286.9606 | 303.9855 | 280.2431 | 293.4608 | 288.2985 |
| 5 | 307.7049 | 300.8550 | 306.9971 | 286.4089 | 291.2267 | 278.8297 | 292.4203 | 283.3925 |
| 6 | 308.4667 | 306.6445 | 303.6845 | 280.5672 | 303.1159 | 289.8373 | 286.3155 | 285.7833 |
| 7 | 308.4288 | 302.0114 | 308.0988 | 281.0041 | 294.3097 | 286.9948 | 272.9784 | 277.9623 |
| 8 | 307.4117 | 306.6916 | 304.3439 | 287.5824 | 300.7196 | 273.4283 | 292.9414 | 280.7263 |
| 9 | 308.8324 | 305.8378 | 308.0631 | 289.3183 | 291.1644 | 278.3319 | 286.2434 | 281.9761 |
| 10 | 308.1411 | 307.5795 | 302.0633 | 279.7089 | 291.7140 | 281.9227 | 288.8347 | 268.1510 |
| Best | 308.8324 | 307.5795 | 308.2725 | 289.3183 | 303.9855 | 289.8373 | 300.8698 | 289.9782 |
| Rank-1 | 1 | 3 | 2 | 8 | 4 | 7 | 5 | 6 |
| Worst | 307.4117 | 300.8550 | 302.0633 | 279.7089 | 291.1644 | 273.2014 | 272.9784 | 268.1510 |
| Rank-2 | 1 | 3 | 2 | 5 | 4 | 6 | 7 | 8 |
| Average | 308.2496 | 304.7252 | 305.5200 | 283.9710 | 297.6146 | 280.5767 | 290.3283 | 281.8569 |
| Rank-3 | 1 | 3 | 2 | 6 | 4 | 8 | 5 | 7 |
| SD | 0.5292 | 2.4531 | 2.4890 | 3.3965 | 5.0304 | 5.2627 | 7.5236 | 6.0933 |
| Rank-4 | 1 | 2 | 3 | 4 | 5 | 6 | 8 | 7 |
| CV | 0.0017 | 0.0081 | 0.0081 | 0.0120 | 0.0169 | 0.0188 | 0.0259 | 0.0216 |
| Rank-5 | 1 | 2 | 2 | 4 | 5 | 6 | 8 | 7 |
| Overall rank | 1 | 3 | 2 | 5 | 4 | 6 | 6 | 8 |

by Eqs. (7)–(9).

$$S_{1,t+1} = S_{1,t} + Q_{1,t} - Re_{1,t} - Sp_{1,t} - Loss_{1,t} \qquad (7)$$

$$S_{2,t+1} = S_{2,t} + Q_{2,t} - Re_{2,t} - Sp_{2,t} - Loss_{2,t} \qquad (8)$$

$$S_{3,t+1} = S_{3,t} + Q_{3,t} + Sp_{1,t} + Sp_{2,t} + Dez_{1,t} + Dez_{2,t} - Re_{3,t} \\ - Sp_{3,t} - Loss_{3,t} \qquad (9)$$

- Constraints of decision variables

$$S_{min\,i} \leq S_{i,t} \leq S_{max\,i} \qquad (10)$$

$$Re_{min\,i,t} \leq Re_{i,t} \leq Re_{max\,i,t} \qquad (11)$$

- Penalty function related to the volume of reservoirs:

$$Penalty1_i = \begin{cases} \sum_{t=1}^{T}\sum_{i=1}^{3} \left(\frac{S_{i,t}-S_{min\,i}}{S_{min\,i}}\right)^2 \text{if } S_{i,t} < S_{min\,i} \\ \sum_{t=1}^{T}\sum_{i=1}^{3} \left(\frac{S_{i,t}-S_{max\,i}}{S_{max\,i}}\right)^2 \text{if } S_{i,t} > S_{max\,i} \\ 0 \text{ if } S_{i,t} \geq S_{min\,i} \text{ and } S_{i,t} \leq S_{max\,i} \end{cases} \qquad (12)$$

where, $Sp_{i,t}$ is the amount of water overflowing from reservoir $i$ in month $t$, $S_{i,t}$ is the storage of reservoir $i$ at the beginning of period $t$, $S_{i,t+1}$ is the storage of reservoir $i$ at the end of period $t$, $S_{max\,i}$ is the maximum storage of reservoir $i$, $S_{min\,i}$, the minimum storage of reservoir $i$, $Q_{i,t}$, the inflow to reservoir $i$ in the month $t$, $Loss_{i,t}$, the amount of losses in reservoir $i$ in month $t$, which is the loss from the reservoir in the form of evaporation and considering the relationship between the reservoir surface and volume, it is calculated based on the Eq. (5) where $A_{i,t}$ is the surface area of reservoir $i$ in period $t$ (square kilometers), $Ev_{i,t}$ is the evaporation rate from reservoir $i$ in period $t$ (m) and $a_i$, $b_i$, and $c_i$ are the constant coefficients of converting the storage of the reservoir $i$

to its corresponding level at the beginning of the same period. $Dez_{i,t}$ is the environmental demand of the reservoir $i$ in month $t$, $Re_{min\,i,t}$ is the minimum release of reservoir $i$ in month $t$, and $Re_{max\,i,t}$, $t$ is the maximum release of the $i$th dam in month $t$.

The next constraint is related to the maximum monthly withdrawal from the aquifer. The upper limit of monthly withdrawal from an aquifer is equal to the total monthly demands of the studied areas in each month of the 19-year period. The other constraints that should be applied are related to groundwater level control. Equation (13) indicates the maximum groundwater level in each area, which should not exceed the plant root zone (1.5 m from the ground). Equation (14) states that groundwater level drop at the end of the operation period should not exceed the maximum allowable drop in the operation period.

$$L_{i,t} \geq 1.5\, i=1,2,3, t=1,2,\ldots,T \qquad (13)$$

$$\sum_{t=1}^{T} \Delta L_{i,t} \leq \Delta L total_{max\,i}, i=1,2,3 \qquad (14)$$

where $L_{i,t}$ is the groundwater depth, i.e., the distance between the well opening and aquifer level in the $i$th region and $t$th month, $\Delta L_{i,t}$ is the change in groundwater level in the $t$th month compared to the $t$-$1$th month in the $i$th region and $\Delta L\_total_{max\,i}$ is the upper limit of drop or lower limit of allowable improvement of the groundwater level at the end of the planning period. Values are selected in such a way that the groundwater level remains almost constant over the 19-year period. Finally, if any of the constraints are not satisfied, the penalty value is added to the objective function according to Eq. (15):

$$Penalty2_i = \sum_{t=1}^{T}\sum_{i=1}^{3} R_i \times \Delta_{i,t} \qquad (15)$$

where $R_i$ is the penalty coefficient in the $i$th region, the appropriate value of which is obtained for each constraint according to the importance of its satisfaction and performing trials and errors in the optimization

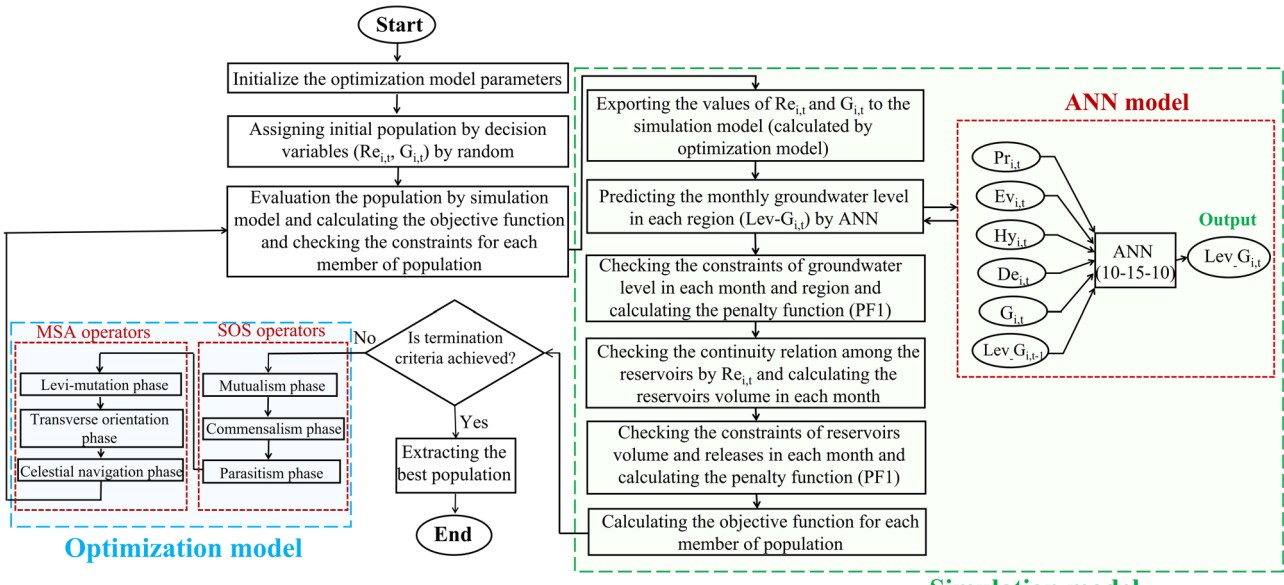

**Fig. 5 | Flowchart of the mechanism of SOS-MSA-ANN simulation-optimization model.** The left box (blue background) demonstrates the optimization model, which was obtained by hybridizing the SOS and MSA algorithms. The right box (green background) shows the artificial intelligence-based simulation model that predicts the groundwater level. The middle part shows the main procedure of the second-level simulation-optimization model.

model, and $\Delta_i$ indicates the extent of exceeding the allowed values in the $i$th region and $t$th month in Eqs. (13) and (14).

## Simulation-optimization model

After the separate development of the simulation (ANN) and optimization (SOS-MSA) models, it is now possible to generate a simulation-optimization model by hybridizing them together. The produced simulation-optimization models (SOS-ANN, MSA-ANN, and SOS-MSA-ANN) can be used for the optimization of the conjunctive operation of surface and groundwater resources in a way that the groundwater level in the study areas reaches a balance. They can be executed by determining the initial parameters of the proposed simulation-optimization models. Supplementary Table S2 represents the parameter setting values for the developed models. These values were obtained by sensitivity analysis. The optimization model should comply with the constraints imposed on the state and decision variables. In this study, groundwater level fluctuation is considered as the state variable, and the amount of allocation of surface and groundwater resources to the demands is the decision variable. First, the optimization model generates decision variables and then, transfers them to the objective function and constraint program. In this program, the inputs are transferred to the ANN (simulation model) to estimate the groundwater level. Next, the simulation model responds to the system behavior based on the optimization model inputs, calculates the groundwater level, and finally, returns it to the optimization model. Constraints are controlled using the penalty method. If the generated solutions can not satisfy the constraints and do not obtain the minimum objective function value, this process continues, so that the optimization algorithm generates new decision variables and sends them to the simulation model. This cycle continues until it reaches the maximum number of iterations and converges to the global or near-global optimum. The flowchart of the developed simulation-optimization model is shown in Fig. 5.

## Performance indicators of water resources systems

Water resources planning and management policies aim to reduce the impact of policies that negatively affect water resources systems both in the current and future situations and to develop policies that have positive socioeconomic, environmental, political, and legal effects on the system. Therefore, there should be some indicators for measuring the system performance to evaluate and compare conditions of water resources systems under different management policies and programs (scenarios)[27]. Evaluating the operation policies is the last and the most important step in utilizing the optimization and simulation models during the operation of water resources systems. In this research, four indicators of reliability, vulnerability, resiliency, and sustainability were used to evaluate the investigated models in the conjunctive operation of surface and groundwater resources.

$$Rel = \left(1 - \frac{NDe_f}{T}\right) \times 100' ND_{ef} = Number\,of\,(De_t > Re_t) \quad (17)$$

where, $Nde_f$ is the overall number of failures during the operation period, $De_t$ is the demand value in period $t$, $Re_t$ is the output in period $t$, and $Rel$ is the system reliability during the operation period. The greater the value of this parameter, the higher the system reliability[28].

$$Vul = \max\left\{\frac{(De_t - Re_t)}{De_t}\right\} \times 100, t = 1, 2, \ldots, T \quad (18)$$

where, $Vul$ denotes the system's vulnerability, $Re_t$ is the output in period $t$, $De_t$ is the demand value in period $t$, and $T$ is the total number of operation periods[28].

$$Res = \frac{\overset{T}{\underset{t=1}{N}} (Def_{t+1} = 0 \mid Def_t > 0)}{\overset{T}{\underset{t=1}{N}} (Def_t > 0)} \times 100, t = 1, 2, \ldots, T \quad (19)$$

where $Res$ denotes the system's resiliency, $\overset{T}{\underset{t=1}{N}} ()$ is the number of occurrences of the condition in parentheses and $Def_t$ is the shortage in period $t$[28].

$$SI = \{Rel \times Res \times (1 - Vul)\}^{1/3} \quad (20)$$

The sustainability index (SI) denotes system performance criteria in a generic index to facilitate comparison and decision-making between different scenarios based on performance indicators of water resources systems[27].

## Data availability
Source data are provided with this paper.

## Code availability
Codes are provided as Supplementary Code 1. The "main_MSA_SOS_hybrid.m" is the main code of the developed simulation-optimization model that was scripted in the programming panel of MATLAB software. The other files include the sub-routines for the main code.

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

## Acknowledgements

We are grateful to Iran National Science Foundation (INSF) and Shahid Bahonar University of Kerman for financial support (Project Code: 4005881). Also, we acknowledge the Research and Technology Institute of Plant Production of Shahid Bahonar University of Kerman.

## Author contributions

S.A. prepared the original draft, collected the data, designed the model, and carried out the simulations. M.R.M. prepared the revisions, analyzed the data, proofread the manuscript, and contributed to the interpretation of the results. M.Z.K. conceived the original idea, reviewed the manuscript, and supervised the project. All authors reviewed the results and approved the final version of the manuscript.

## Competing interests

The authors declare no competing interests.
