## [Peer Review File · Nature Communications]

An artificial intelligence-based model for optimal conjunctive operation of surface and groundwater resourcesREVIEWER COMMENTS

Reviewer #1 (Remarks to the Author):

This manuscript presents a hybrid SOS-MSA algorithm to optimize conjunctive use of surface and ground water resources of Halilrood basin in Iran. Sustainability of water resources systems including reliability, resilience, and vulnerability is evaluated.

In general, the manuscript is hardly readable and understandable. The method of analysis is not clearly presented in the manuscript. Major findings (and advancements over previous studies) are not clearly formulated. Mistakes/errors in the text of manuscript are so numerous that I just do not see a sense to mention all of them.

Introduction: Authors should be informed about the results of studies which they cited in this section. It's not enough to just cite the works. In addition, what the "problem statement" is exactly must be explained via the scientific way.

Study area: The authors should present complete descriptions of the interaction of surface and ground water sources during the simulation period.

Methodology: The objective function and constraints used are very simple and it cannot show the superiority of the proposed optimization method compared to other methods.

Results and Discussions: The main results of the study should be given in comparison with previous studies.

Finally, the manuscript does not have the research innovation of a valuable Nature Communications Journal. In conclusion, I see this manuscript more appropriate for an applied journal.

Reviewer #2 (Remarks to the Author):

Comments:

The manuscript addresses the optimal conjunctive operation of surface and groundwater resources using a Novel Hybrid SOS-MSA-ANN method for analyzing management scenarios. The paper is well written and structured.

A discussion section, which should be based on references findings of other authors, is lacking citations.

Please separate the Results and discussion section and provide a proper and thorough discussion of the results.

Specific comments:

1.Line 12: Can the authors change "i.e" to "For example" or rephrase the sentence?

2.What is the problem of the research? Agricultural water shortage?

3.Authors mentioned that the study area is mostly irrigated. What is the main source of irrigation water? River or groundwater?

4.Figure 9: Can the authors add the baseline of the demand deficit to the simulated demand deficits.

Reviewer #3 (Remarks to the Author):

A novel simulation-optimization model, namely SOS-MSA-ANN model, was developed for the integrated management of water resources under the scenario of conjunctive operation of surface and groundwater resources by connecting an artificial neural network based on radial basis function (ANN-RBF) simulation model to the proposed hybrid SOS-MSA algorithm. The results are quite good at tackling highly conventional complex water management in optimizing conjunctive use by the development of a combination of a set of metaheuristics. The applicability of the method has been demonstrated through real-field situations in Iran. Data soundness is also ok. I consider that the paper can be published as such though I have a comment on the logic of data-driven models only for these purposes but there is enough material on this paper to be published. I suggest to the authors for another paper to combine conceptual models (groundwater and surface water) with data and metaheuristics or a combination for optimization purposes. A benchmark of data-driven models and conceptual models is welcome and highly desirable.

Reviewer #4 (Remarks to the Author):

1. The paper must be thoroughly English edited before it is accepted for publication.
2. The paper presents the application of several evolutionary and machine-learning algorithms to manage the surface water and groundwater resources of an arid basin in Iran. There is novelty in the use of such algorithms and their application.
3. I question the wisdom and appropriateness of testing the 8 proposed MSA structures with the classic four-reservoir system by Murray and Yakowitz (1979) because that system does involve groundwater resources, only surface water resources, whereas the reviewed paper involves surface water resources and groundwater resources. Therefore, if the MSA structures are tested, they should be tested with a system that involves surface water and groundwater resources, not with a simple four-reservoir system.
4. Line 172: use hot summers instead of "scorching summers".
5. Table 1: I believe "evaporation" is in fact "potential evapotranspiration". Notice that in the Bath region the average annual (?) precipitation equals 20.05 mm, whereas the "evaporation" equals 206.33 mm (much larger than the precipitation). This suggests the authors mean "potential evapotranspiration". Also, it is important to specify that the values listed in Table 1 represent average annual quantities.
6. The groundwater level units in Table 1 is MSAL. What does MSAL mean? It is customary to use MASL, which means meters above mean sea level.
7. Line 272: replace "getting stuck in local optima" with "being trapped in local optima".
8. Line 330: replace "global optimum value" with "global optimal value".
9. Table 1: replace "demands" with "water demand".
10. Objective function, equation (3): I believe the summation must be over all months ($i=1, 2, \dots, 12$), and over all regions ($k=1, 2, 3$). Therefore, the objective function must have two summation operators, not just one for months, unless each region is managed independently of the others as suggested by the results listed in Tables 4 and 5.
11. Provide references for the definitions of reliability, resiliency, vulnerability, and sustainability.
12. Line 434: how were determined the parameters of the proposed SOS-MSA-ANN simulation-optimization model? how many parameters are there? what are their values?
13. I recommend accepting the paper after major revision and re-review.

Responses to Reviewer#1 comments

Comment 1: This manuscript presents a hybrid SOS-MSA algorithm to optimize conjunctive use of surface and ground water resources of Halilrood basin in Iran. Sustainability of water resources systems including reliability, resilience, and vulnerability is evaluated.

In general, the manuscript is hardly readable and understandable. The method of analysis is not clearly presented in the manuscript. Major findings (and advancements over previous studies) are not clearly formulated. Mistakes/errors in the text of manuscript are so numerous that I just do not see a sense to mention all of them.

Response # First, we would like to extend our sincere gratitude to you for the thorough consideration and critical comments that have indeed helped us improve the quality of the manuscript. We have revised the manuscript according to the constructive comments. We hope that our revisions have improved the paper to a level of your satisfaction.

Second, we deeply attempted to improve the English usage. We got help from an English Editing Service Institute as well as we used the premium version of Grammarly to fix all mistakes/errors in the text.

Third, we deeply tried to improve the methodology by adding more detailed formulae and adding more description and flowcharts with the purpose of clearing the methodology.

Fourth, we added a new section, Discussion, to the revised manuscript. In this section we compared the major findings of our study (and its advancements) with the previous studies

Comment 2: Introduction: Authors should be informed about the results of studies which they cited in this section. It's not enough to just cite the works. In addition, what the "problem statement" is exactly must be explained via the scientific way.

Response # In the revised version of manuscript, we made a new literature review and tried to present the results of previous studies in a clearer way. In addition, we deeply attempted to highlight the problem statement and research novelty (please see the revised Introduction). The purpose of this study was to optimize the conjunctive operation of surface and groundwater resources in the Halilrood basin, in such a way that the water deficit in the study areas is minimized (by the optimized water supply) and at the same time, the amount of water withdrawal from the groundwater resources remains at a sustainable condition, so that the groundwater level reach a balance during the 19-year period. In other word, by the management scenarios produced by the utilized models, while a higher percentage of the basin's demands has been met, the groundwater level has become more stable, and in other words, the sustainable operation of groundwater resources was taken place.

Comment 3: Study area: The authors should present complete descriptions of the interaction of surface and ground water sources during the simulation period.

Response # As you rightly pointed out, the conjunctive operation of surface and groundwater resources can only be optimized with a detailed understanding of the relationship and interaction of these resources. When the water extraction from the surface resources is prioritized in the optimization model, the flood control is simultaneously taken into account and the problem of sharp drop of groundwater level due to the excessive withdrawal of water from the

groundwater resources is taken into consideration. In fact, the simulation-optimization process is done with a comprehensive look at the surface and groundwater resources in the form of a system.

Comment 4: Methodology: The objective function and constraints used are very simple and it cannot show the superiority of the proposed optimization method compared to other methods.

Response # In the revised manuscript, we modified the objective function and presented more details of the optimization constraints. It should be noted that this problem has 1338 decision variables (3 regions * 2 sources of water (surface and groundwater) * 223 months of operation =1338 decision variables), indicating the high complexity of the problem. Therefore, this is not a simple problem.

Comment 5: Results and Discussions: The main results of the study should be given in comparison with previous studies.

Response # per your comment, we compared the results of this work with the results of the previous studies. Accordingly, we added a new section, Discussion, to the revised manuscript, and provided the comparisons with the others in this section, as you recommended.

Comment 6: Finally, the manuscript does not have the research innovation of a valuable Nature Communications Journal. In conclusion, I see this manuscript more appropriate for an applied journal.

Response # In the revised manuscript, we deeply tried to highlight the novelty of this work (please see the revised Introduction). The purpose of this study was to optimize the conjunctive operation of surface and groundwater resources in the Halilrood basin in such a way that the water deficit in the study areas is minimized (by the optimized water supply) and at the same time, the amount of water withdrawal from the groundwater resources remains at a sustainable condition, so that the groundwater level reach a balance during the 19-year period. In other word, by the management scenarios produced by the utilized models, while a higher percentage of the basin's demands has been met, the groundwater level has become more stable, and in other words, the sustainable operation of groundwater resources was taken place.

Responses to Reviewer#2 comments

Comment 1: The manuscript addresses the optimal conjunctive operation of surface and groundwater resources using a Novel Hybrid SOS-MSA-ANN method for analyzing management scenarios. The paper is well written and structured.

Response # First, we would like to extend our sincere gratitude to you for the thorough consideration and critical comments that have indeed helped us improve the quality of the manuscript. We have revised the manuscript according to the constructive comments. We hope that our revisions have improved the paper to a level of your satisfaction.

Comment 2: A discussion section, which should be based on references findings of other authors, is lacking citations.

Response # In the revised manuscript, we added a new section, Discussion, and provided the comparisons with the others in this section, as you recommended.

Comment 3: Please separate the Results and discussion section and provide a proper and thorough discussion of the results.

Response # In the revised manuscript, we separated the Results with the Discussion, as you recommended.

Comment 4: Line 12: Can the authors change “i.e” to “For example” or rephrase the sentence?

Response # we rephrased the sentence as you suggested.

Comment 5: What is the problem of the research? Agricultural water shortage?

Response # The demands of the studied areas include domestic, agriculture, industry, and environmental demands. The purpose of this research was to minimize the deficits in meeting these demands (please see the objective function) with the priority of meeting the domestic and environmental demands. To supply the water for these demands, both surface water and groundwater resources were considered. This study aimed at optimizing the conjunctive operation of surface and groundwater resources in the Halilrood basin in such a way that the water deficit in the study areas is minimized (by the optimized water supply) and at the same time the amount of water withdrawal from the groundwater resources remains at a sustainable condition, so that the groundwater level reach a balance during the 19-year period. In other word, by the management scenarios produced by the utilized models, while a higher percentage of the basin's demands has been met, the groundwater level has become more stable, and in other words, the sustainable operation of groundwater resources was taken place. It should be noted that, in the revised manuscript, we deeply attempted to highlight the problem statement and research novelty (please see the revised Introduction).

Comment 6: Authors mentioned that the study area is mostly irrigated. What is the main source of irrigation water? River or groundwater?

Response # Kindly, considering the hundreds of unauthorized wells with uncertain amounts of water withdrawal, it is difficult to exactly determine the main source of water withdrawal in real conditions (non-optimal condition). In fact, both the groundwater and surface water withdrawals occur in the studied regions. For example, in Baft and Rabor regions, the water supply is mostly done from surface water sources, but in Jiroft region, the focus is more on the groundwater resources. This is exactly why we seek to optimize the conjunctive operation of surface and groundwater resources in these regions. In the optimized condition, to balance withdrawal from aquifer, we developed the simulation-optimization model in such a way that first the water extraction should be done from the groundwater and then from the surface water.

Comment 7: Figure 9: Can the authors add the baseline of the demand deficit to the simulated demand deficits.

Response # Done. Please see Figure (9) in the revised manuscript. Thank you for your valuable recommendation.

Responses to Reviewer#3 comments

Comment 1: A novel simulation-optimization model, namely SOS-MSA-ANN model, was developed for the integrated management of water resources under the scenario of conjunctive operation of surface and groundwater resources by connecting an artificial neural network based on radial basis function(ANN-RBF) simulation model to the proposed hybrid SOS-MSA algorithm. The results are quite good at tackling highly conventional complex water management in optimizing conjunctive use by the development of a combination of a set of metaheuristics. The applicability of the method has been demonstrated through real-field situations in Iran. Data soundness is also ok.

Response # First, we would like to extend our sincere gratitude to you for the thorough consideration and critical comments that have indeed helped us improve the quality of the manuscript. We have revised the manuscript according to the constructive comments. We hope that our revisions have improved the paper to a level of your satisfaction.

Comment 2: I consider that the paper can be published as such though I have a comment on the logic of data-driven models only for these purposes but there is enough material on this paper to be published. I suggest to the authors for another paper to combine conceptual models (groundwater and surface water) with data and metaheuristics or a combination for optimization purposes. A benchmark of data-driven models and conceptual models is welcome and highly desirable.

Response # Thank you for your valuable suggestion. We will consider your precious suggestions in our upcoming paper.

Responses to Reviewer#4 comments

Comment 1: The paper must be thoroughly English edited before it is accepted for publication. The paper presents the application of several evolutionary and machine-learning algorithms to manage the surface water and groundwater resources of an arid basin in Iran. There is novelty in the use of such algorithms and their application.

Response # First, we would like to extend our sincere gratitude to you for the thorough consideration and critical comments that have indeed helped us improve the quality of the manuscript. We have revised the manuscript according to the constructive comments.
In the revised manuscript, we deeply attempted to fix all the grammatical, punctuation and spelling errors. We got help from an English Editing Service Institute to improve the English usage. We hope that our revisions have improved the paper to a level of your satisfaction.

Comment 2: I question the wisdom and appropriateness of testing the 8 proposed MSA structures with the classic four-reservoir system by Murray and Yakowitz (1979) because that system does involve groundwater resources, only surface water resources, whereas the reviewed paper involves surface water resources and groundwater resources. Therefore, if the MSA

structures are tested, they should be tested with a system that involves surface water and groundwater resources, not with a simple four-reservoir system.

Response # Thank you so much for your valuable comment. To investigate the effect of these 8 structures on the performance of MSA algorithm and to identify its strongest and most effective operators, a benchmark problem was needed. As you know, most of the benchmarks are mathematical test function such as Rosenbrock, Zitzler–Deb–Thiele's functions, etc. To the author's knowledge, the 4-reservoir and 10-reservoir benchmark problem is the only engineering benchmarks in water resources optimization. Based on our extensive research, no such function has been found that both take into account the surface and groundwater problems simultaneously. Accordingly, the four-reservoir benchmark problem, as the closest test problem to the problem of conjunctive operation of surface and groundwater resources, was used (Murray and Yakowitz, 1979).

Comment 3: Line 172: use hot summers instead of "scorching summers".

Response # Done

Comment 4: Table 1: I believe "evaporation" is in fact "potential evapotranspiration". Notice that in the Bath region the average annual (?) precipitation equals 20.05 mm, whereas the "evaporation" equals 206.33 mm (much larger than the precipitation). This suggests the authors mean "potential evapotranspiration". Also, it is important to specify that the values listed in Table 1 represent average annual quantities.

Response # In the study regions, even though the amount of precipitation is less than one third of the world average, but the actual evaporation is about 6 times the world average. In other words, the amount of evaporation in these arid areas is much higher than the amount of precipitation. In these regions, the stored water. In these areas, the water stored behind the dams (reservoirs) is often caused by seasonal floods. The values mentioned in the Table are indeed the evaporation from the standard Class A evaporation pan. In the revised manuscript, we specified that the values listed in Table 1 represent average annual quantities. Please see the table caption.

Comment 5: The groundwater level units in Table 1 is MSAL. What does MSAL mean? It is customary to use MASL, which means meters above mean sea level.

Response # Corrected. Thank you for your careful review.

Comment 6: Line 272: replace "getting stuck in local optima" with "being trapped in local optima".

Response # Done.

Comment 7: Line 330: replace "global optimum value" with "global optimal value".

Response # Done.

Comment 8: Table 1: replace "demands" with "water demand".

Response # Done.

Comment 9: Objective function, equation (3): I believe the summation must be over all months ($i=1, 2, \dots, 12$), and over all regions ($k=1, 2, 3$). Therefore, the objective function must have two summation operators, not just one for months, unless each region is managed independently of the others as suggested by the results listed in Tables 4 and 5.

Response # In the revised manuscript, the objective function has been corrected and some modifications were carried out on the constraints and utilized formulas.

Comment 10: Provide references for the definitions of reliability, resiliency, vulnerability, and sustainability.

Response # In the revised manuscript we added “Hashimoto et al., 1982” and “Sandoval-Solis et al., 2011” as the references of these criteria.

Comment 11: Line 434: how were determined the parameters of the proposed SOS-MSA-ANN simulation-optimization model? how many parameters are there? what are their values?

Response # The parameters of the utilized models were determined by the sensitivity analysis. Based on your comment, we added Table (3) to the revised manuscript to present the values of setting parameters of each model.

Comment 12: I recommend accepting the paper after major revision and re-review.

Response # Thank you so much for the time and energy you spent on the reviewing this article and thank you again for your constructive comments and valuable recommendations.

At the end, we again thank the Editor(s) and the Reviewers for their constructive comments and valuable suggestions. We found them quite useful as we approached our revision. We hope that our revision has improved the paper to a level of your satisfaction.

REVIEWER COMMENTS

Reviewer #1 (Remarks to the Author):

I have read the revised manuscript, together with the earlier reviews and the authors' response. The authors have addressed all the review comments and made appropriate modifications to the manuscript. I recommend acceptance of the manuscript for publication.

Reviewer #2 (Remarks to the Author):

The authors have addressed all the comments satisfactorily.

Reviewer #4 (Remarks to the Author):

1. I recommend major revision.
2. The paper presents a novel simulation-optimization method for surface water / groundwater use, employing evolutionary algorithms and ANN;
3. The revised paper is a major improvement relative to the first version; however:
4. English editing is required, the paper has many grammatical errors, too many for me to outline and correct in this review.
5. Lines 379-380 define the groundwater exploitation (it should be "groundwater withdrawal") $G_{i,t}$, but $G_{i,t}$ does not appear in equation (3);
6. Equation (3) involves $Penalty1_i$ and $Penalty2_i$, but these are not defined in lines 376-380 explaining the variables that appear in equation (3).
7. In fact, $Penalty1_i$ is defined in equation (12), but $Penalty2_i$ is not defined in the paper.
8. Therefore, the mathematics of this paper are incomplete and one cannot reproduce or verify its results.
9. A major revision is recommended.

Responses to Reviewer#1 comments

Comment 1: I have read the revised manuscript, together with the earlier reviews and the authors' response. The authors have addressed all the review comments and made appropriate modifications to the manuscript. I recommend acceptance of the manuscript for publication.

Response # Thank you so much.

Responses to Reviewer#2 comments

Comment 1: The authors have addressed all the comments satisfactorily.

Response # Thank you so much.

Responses to Reviewer#4 comments

Comment 1: I recommend major revision. The paper presents a novel simulation-optimization method for surface water / groundwater use, employing evolutionary algorithms and ANN; The revised paper is a major improvement relative to the first version; however: English editing is required, the paper has many grammatical errors, too many for me to outline and correct in this review.

Response # In the re-revised version, we got help from a native English editor to fix all grammatical errors in the text. Please see the English editing certificate.

Comment 2: Lines 379-380 define the groundwater exploitation (it should be "groundwater withdrawal") $G_{i,t}$, but $G_{i,t}$ does not appear in equation (3)

Response # Based on your valuable comment, we replaced the term "groundwater exploitation" with the "groundwater withdrawal". As seen, the G_i was mentioned in the ($Supply_{i,t} = G_{i,t} + Re_{i,t}$) as in-line (Line 378). In addition, we presented the parameters in the text in the same order as mentioned in Eq. (3). Please see the revised manuscript.

Comment 3: Equation (3) involves $Penalty1_i$ and $Penalty2_i$, but these are not defined in lines 376-380 explaining the variables that appear in equation (3). In fact, $Penalty1_i$ is defined in equation (12), but $Penalty2_i$ is not defined in the paper. Therefore, the mathematics of this paper are incomplete, and one cannot reproduce or verify its results. A major revision is recommended.

Response # As seen, the $Penalty1_i$ was defined in Eq. (12) and the $Penalty2_i$ was defined in Eq. (15). For further clarification, we made some modifications on the mathematical explanations and added the below sentence to this section (Lines 380~382)

" $Penalty1_i$ and $Penalty2_i$ are the penalty functions related to the volume of reservoirs (Eq. 12) and groundwater level (Eq.15), respectively."

At the end, we again thank the Editor(s) and the Reviewers for their constructive comments and valuable suggestions. We found them quite useful as we approached our revision. We hope that our revision has improved the paper to a level of your satisfaction.

REVIEWERS' COMMENTS

Reviewer #4 (Remarks to the Author):

The authors have revised their article and it is acceptable in its present form.
I do not have any other recommendations for improvement.